# Deep Reinforcement Learning for the Detection of Abnormal Data in Smart Meters

**DOI:** 10.3390/s22218543

**Published:** 2022-11-06

**Authors:** Shuxian Sun, Chunyu Liu, Yiqun Zhu, Haihang He, Shuai Xiao, Jiabao Wen

**Affiliations:** 1Marketing Service Center, State Grid Tianjin Electric Power Company, Tianjin 300120, China; 2School of Electrical and Information Engineering, Tianjin University, Tianjin 300072, China

**Keywords:** deep reinforcement learning, smart meters, *Q*-learning

## Abstract

The rapidly growing power data in smart grids have created difficulties in security management. The processing of large-scale power data with the use of artificial intelligence methods has become a hotspot research topic. Considering the early warning detection problem of smart meters, this paper proposes an abnormal data detection network based on Deep Reinforcement Learning, which includes a main network and a target network composed of deep learning networks. This work uses the greedy policy algorithm to find the action of the maximum value of *Q* based on the *Q*-learning method to obtain the optimal calculation policy. It also uses the reward value and discount factor to optimize the target value. In particular, this study uses the fuzzy c-means method to predict the future state information value, which improves the computational accuracy of the Deep Reinforcement Learning model. The experimental results show that compared with the traditional smart meter data anomaly detection method, the proposed model improves the accuracy of meter data anomaly detection.

## 1. Introduction

With the development of a new generation of information technologies, the power industry is undergoing tremendous changes. The smart grid plays an important role in promoting the sustainable development of the world economy and society. It can optimize the energy structure, improve the utilization rate of energy, improve the security and stability of the power grid, and promote technology innovation in related fields as well as realize a two-way interaction between the power grid and its users. It can be seen that the comprehensive construction of the smart grid has become the main direction of the future development of the grid. However, smart metering faults have caused serious losses to customers and power companies. For example, abnormal readings of smart meter data have caused financial losses to power companies. In addition, there are other faults that cause problems for the power company’s work. Considering the problem of abnormal smart meter data detection, this paper proposes an intelligent method to reduce the amount of work resulting from smart metering faults.

With the improvement of computing power and the emergence of high-performance devices, machine learning has developed rapidly, and this has produced many kinds of algorithms. Among them, a promising machine learning algorithm called Reinforcement Learning (RL) [1,2] has attracted the attention of researchers. RL is closer to human learning and actively guides the agent to learn strategies in the process of interacting with the external environment in order to obtain maximum benefits and achieve target needs. RL fundamentally breaks the previous traditional thinking of the DL processing data by building learning models and engaging in model training and testing, and it solves sequence decision-making problems through value functions, strategies, and other perspectives. It has been widely used in simulation modeling, robot control, games, and other fields. Since RL is more focused on exploring strategies to solve problems, it has been unable to handle the current, overly complex real-world tasks. In order to solve the above problems, Google’s DeepMind team proposed one of the most promising architectures, namely Deep Reinforcement Learning (DRL) [3,4,5,6], which combines the decision-making ability of RL and the perception ability of DL. It is not only capable of processing high-dimensional continuous action space data, but it can also directly calculate the original input to output data.

Therefore, this paper proposes an abnormal detection method for smart meter data based on Deep Reinforcement Learning. The method includes two networks: The first is the current network, which uses the neural network model to reconstruct the power characteristics and then performs prediction and classification; the second is the target network, which inputs the power characteristics of the next stage and uses the unsupervised learning algorithm, i.e., the fuzzy c-means (FCM), to cluster the power characteristics and output the cluster labels. In addition, we use the target neural network to calculate the value of the *Q* function, which represents the expectation of the total reward that the agent can obtain in the future after taking action *a* in state *s* in the Q-learning algorithm. The experimental results show that the proposed method achieves better results in the abnormal detection of smart meter data. The proposed method prevents the unnecessary economic loss and harm caused by the abnormal power consumption of users, and it greatly improves the rate of detection of abnormal users by power companies. The main contributions of this paper are as follows:(1)In view of the problem of massive, diverse, and complex factors influencing electricity consumption data from electricity meters, this paper proposes an abnormal data detection model for smart meters based on Deep Reinforcement Learning.(2)In this paper, the FCM algorithm is used to realize semi-supervised learning in the DQN network, to predict the sample state in the next moment through the FCM algorithm, and then to predict its *Q* value through the target network.(3)The method proposed in this paper is analyzed and tested on a real user power consumption dataset. The proposed method can significantly improve detection accuracy and speed, and the model shows strong generalization ability and applicability.

## 2. Related Work

### 2.1. Smart Grid Security Technologies

In improving the operation efficiency of the power system, increasing the power supply, enhancing the service quality of the application, and promoting energy savings and emissions reduction, countries around the world have used the development, improvement, and reform of the power supply industry as their primary strategy. The popularization and application of smart energy meters and electricity consumption information collection systems, along with the combination of hardware and software, have greatly benefited power supply companies and power users. Power users can understand their power consumption in real time through this system. The system can monitor the electricity consumption dynamics of the area at any time, promote a two-way dialogue between electricity users and power companies, make the distribution of electricity resources more reasonable, and also provide a solid data foundation and technical support for the smooth progress of electricity theft prevention work.

Reference [7] proposed an access control method—from identity establishment and control to identification in the case of protection problems in advanced measurement systems as well as data acquisition and monitoring in control systems; the method was mainly used to defend against attacks on the system. Reference [8] carried out a security analysis and research on the related information security problems of smart grids, mainly from the generation, transmission, and processing of data. Reference [9] adopted the research strategy of data aggregation and used a homomorphic encryption algorithm to prevent data from being leaked during transmission and storage. In reference [10], aiming at the distributed denial of service attack problem in the advanced measurement system, the support vector machine and artificial immune system were combined to identify malicious data existing in the network and effectively detect the possible attack behavior in it. Reference [11] used the method combining K-means, particle swarm, and an improved support vector machine to discriminate the detected abnormal data from the suspected user, and then to detect the suspected user by focusing on solving the abnormal electricity consumption in the smart grid.

### 2.2. Abnormal Detection Method of Smart Meter Data

Considering the data detection problem of smart meters, two main schemes in current research works can be identified. These include the detection method based on data mining and the manual sampling method. The manual sampling method is mainly a method of inspecting electricity theft, which is subjectively conducted by the staff [12]. Hardware facilities and manual participation are two indispensable factors of this detection method. With regard to the hardware equipment [13,14,15] such as the special metering cabinet used to detect electricity stealing behavior, when the characteristic electrical parameters of voltage and current (recorded by the metering cabinet) are significantly different from the normal data, the user will be marked as the suspected object of electricity theft. However, equipping each user with a dedicated metering cabinet will consume a lot of human and financial resources, and in a large environment, the effect is obviously not good and unrealistic. In terms of manual participation, the power supply company needs to regularly dispatch a large number of staff to screen for potential electricity theft areas, but the number of electricity users is large, the screening efficiency is extremely low, and the timeliness and accuracy cannot be guaranteed. In addition, this kind of method cannot accurately and efficiently analyze all kinds of electricity stealing behaviors, and the collected abnormal electricity consumption information is relatively simple. The characteristics of high delay directly leads to high misjudgments and missed judgments, resulting in the ineffectiveness of preventing electricity theft.

The detection methods based on data mining can be divided into unsupervised learning methods and supervised learning methods. The unsupervised learning method identifies suspected objects from a large number of users by comparing users horizontally; this comparison is based on the characteristics of all “power thieves” without additional prompts. In addition, unsupervised learning can well avoid the influence of objective conditions on the judgment of different electricity thieves in the same environment. According to the theory of outliers, some studies [16] have explored a distance-based abnormal detection method for electricity consumption. By analyzing a large amount of data, the outlier boundary was determined so as to achieve the effect of predicting the suspected object of electricity theft. Another study [3] established an unsupervised abnormal detection model based on the local outlier factor algorithm, took electricity as a characteristic index, introduced grid technology, and detected electricity anomaly objects through the LOF algorithm. Another supervised learning method required a large amount of training data, that is, it required a large amount of data from electricity users for training, with known results, so as to determine whether an unknown user was suspected of stealing electricity. Some authors used the one-class SVM classification algorithm to determine the threshold range of normal electricity consumption by analyzing the data of a large number of normal users [4]. At the same time, in order to reduce the rate of false alarms, filtering technology was adopted. Another author [5] proposed an electricity stealing behavior recognition algorithm for users that was based on the semi-supervised learning of the L0 sparse hypergraph, and then constructed an L0 sparse hypergraph model, calculated the weight matrix, looked for the relationship between the points of the hypergraph, and used a small amount of learning with labels. The sample was able to predict the suspected object of electricity theft. The paper by [6] analyzed the suspected factors for user electricity theft and proposed an evaluation model based on an improved genetic optimization neural network. Aiming at the defects of the BP neural network, the genetic algorithm was used to optimize the performance, which effectively prevented a fall into the local optimum.

### 2.3. Deep Reinforcement Learning

In this work, we introduce a reinforcement learning method to realize state prediction with the use of power data. Deep Reinforcement Learning (DRL) is mainly model-free and model-based. Model-free methods include those based on value function and policy gradient, while model-based ones include methods based on search and supervised learning. DRL originated from the Deep *Q*-learning Network (DQN) algorithm proposed by Mnih et al. [17] for solving vision-related control problems. Instead of the action-value table in the traditional *Q*-learning algorithm, the deep Q-network approximates the action-value function as a Convolutional Neural Network (CNN) and is called an action-value network. In addition, the DQN also innovatively uses the experience replay mechanism, utilizing the experience replay memory to store the rewards and states obtained from each interaction with the environment, and to update the parameters of the iterative action value network. An improved version of the DQN [18] employs an isomorphic network to generate an action value as a target, thereby reducing the correlation between the current and target action values and improving the stability of the algorithm. At the same time, researchers can use the same set of training network and algorithm parameters to deal with different complex visual perception tasks in the experiment, which illustrates the versatility of the DQN. Practice has shown that DQNs have surpassed the human level in some non-strategic games [19].

The Deep Reinforcement Learning method based on policy gradient is mainly used to solve various problems that exist in the algorithm based on action value, such as the insufficient processing ability of continuous action. The Actor-Critic algorithm is implemented in combination with the value function approach. On this basis, Silver et al. [20] and Lillicrap et al. [21] proposed the Deep Deterministic Policy Gradient (DDPG) algorithm using the principle of deterministic policy gradient. Experiments have shown that the algorithm is not only stable in the continuous action space, but that the solution speed of the optimization strategy is also faster than the calculation method based on action value.

Deep Reinforcement Learning methods based on search and supervision functions are widely used in action planning problems in games. The AlphaGo Go algorithm [22] combines deep neural networks with a classic heuristic search method, the Monte Carlo Tree Search [23] (MCTS), and facilitates policy search with additional human supervision. The algorithm approximates a value function using the Monte Carlo Tree Search and utilizes a Convolutional Neural Network to evaluate the Go layout based on this value function.

## 3. Method

Considering the problem of abnormal electricity consumption of smart meters, this paper proposes a smart meter data abnormal detection model based on Deep Reinforcement Learning, which combines the *Q*-learning algorithm in Reinforcement Learning and the neural network in deep learning to resolve the abnormal behavior of the meter data.

### 3.1. Overview of the Deep *Q*-Network Model

The *Q*-learning algorithm [24] is a value-based Reinforcement Learning algorithm. The method used by the *Q*-learning algorithm in the iteration is the reward and value function of the state action. The *Q*-learning algorithm first initializes the model and all *Q* values, and the agent performs the action in the current state to obtain the vector $\left(x_{t},u_{t},x_{t+1},u_{t+1}\right)$, using the greedy algorithm to select the action and then, according to Algorithm 1, to update the *Q* value. When the agent reaches the target state, the *Q*-learning algorithm is terminated, and the process of this iteration is completed. Then, the algorithm iterates repeatedly until the *Q* value finally converges and the *Q* learning algorithm is over. The *Q*-learning algorithm is shown in Algorithm 1.
**Algorithm 1** Q-learning algorithm.1:**repeat**2:   each data item for each mini-batch sample3:   using a greedy strategy, choose action $u_{t}$, get reward $r_{t}$, and reach a new state $x_{t+1}$4:   $Q\left(x_{t},u_{t}\right)\leftarrow Q\left(x_{t},u_{t}\right)+\alpha \left[r_{t+1}+\gamma \max Q\left(x_{t+1},u_{t+1}\right)-Q\left(x_{t},u_{t}\right)\right]$5:   $x_{t}\leftarrow x_{t+1}$6:**until** all Q(x,u) reach a state of convergence

This paper transforms the problem so that the *Q*-table updating problem becomes a function fitting problem. The DRN uses the neural network as the approximator of the *Q* function and then uses Reinforcement Learning to adjust the parameters of the neural network of the deep learning model. It subsequently realizes the modeling of the *Q* table, and finally enables the agent to obtain an optimal strategy [25].

The goal of the DQN model is to obtain the optimal *Q* function through the policy function, which reflects the state *s* and the action *a* at each moment. The policy function depends on the state, which is derived by the *Q*-function from the following function:(1)$$F_{policy}\left(s\right)=\mathrm{argmax}\left(Q\left(s,a\right)\right)$$

This paper uses a simple DNN to ensure a positive *Q* value. The network model is trained using the Mean Squared Error (MSE) loss, which is the value of the next state obtained by multiplying the current reward by *Q*, multiplied by a discount factor.

### 3.2. Model Design

DQN has two great advantages—a target neural network and a playback memory unit—which improve the stability of the neural network. Moreover, during the training process, DQN directly inputs the original data into the neural network without adding additional data information. DQN utilizes the off-policy feature of *Q*-learning. The experience replay mechanism in DQN is obtained through the experience data $m=\{s,a,s_{t+1}\}$ stored at each time node. During the training process, DQN reads the sample data through the mini-batch sampling and then uses the gradient descent algorithm to update the deep network parameters. This method effectively utilizes the coupling between data and improves the stability of the model. In this paper, the structure of the initial network in DQN and the target network are set to be the same. The parameter update of the target network is obtained by the *Q* network after N iterations, and its loss function is expressed as follows:(2)$$Loss=\frac{1}{m}\sum_{t=1}^{m}{\left(r_{t}+\gamma \max Q\left(s_{t+1},a_{t+1}\right)-Q\left(s_{t},a_{t}\right)\right)}^{2}$$

Figure 1 shows the training process of the DQN network model, which consists of the main network (MainNet) for predicting the *Q* estimation and the target network (TargetNet) for predicting the *Q* reality; the main network uses the latest parameters. The target network uses the previous parameters, and the actual target *Q* is calculated as follows:(3)$$Q_{target}=r+Q\max\left(s_{t+1},a_{t+1},\theta \right)$$

Moreover, its loss function is
(4)$$Loss\left(\theta \right)=E[Q_{target}-Q\max\left(s_{t},a_{t},\theta \right)].$$

After initializing MainNet and TargetNet, the parameters of MainNet are updated according to the loss function, while those of TargetNet are fixed. After many iterations (epoch = 100), all the parameters of MainNet are copied into TargetNet. During this time, the constant target *Q* value allows the model update to be more stable. Generally, there are noise samples in the dataset. In order to improve the robustness of the model, this study adds the fuzzy c-means (FCM) model to predict the action corresponding to each state when the target network is updated. In this algorithm, the state $s_{t+1}$ is the input, and its sampling distance is used as the centroid vector $u_{t+1}$. Based on the Euclidean distance between *s* and *u*, the prediction samples are divided fuzzily.

The FCM algorithm is the minimization of the objective function consisting of the membership degree of sample data, clustering centers, and sample centers. Suppose that the sample set is $S=\{s_{1},s_{2},s_{3},\ldots ,s_{n}\}$, and find the central $v_{j}\left(j=1,2,\ldots ,c\right)$ of each category. The objective function is then minimized, as seen below:(5)$$J_{m}\left(U,v_{1},v_{2},\ldots ,v_{c}\right)=\sum_{j=1}^{n}\sum_{i=1}^{c}u_{ij}^{m}d_{ij}^{2}\sqrt{b^{2}-4ac}$$
where *n* represents the number of samples in the data set, *m* denotes the fuzzy weighting coefficient, *c* represents the number of clusters, and $u_{ij}$ represents the membership degree of the sample point $x_{i}$ belonging to the *j*th clustering center. $U=\{u_{ij}\}$ represents the membership matrix. $u_{ij}$ has the following constraints:(6)$$\sum_{j=1}^{c}u_{ij}=1,\begin{array}{cc} & \forall i=1,2,\ldots ,n \end{array}$$
(7)$$0<\sum_{i=1}^{n}u_{ij}<n,\begin{array}{cc} & \forall j=1,2,\ldots ,c. \end{array}$$

In Formula (Equation 5), $d_{ij}$ is the Euclidean distance between the ist cluster center and the jst sample point. For (Equation 6) and (Equation 7), Lagrange operators are used to construct the following objective functions:(8)$$\begin{array}{c} J\left(U,v_{1},\ldots ,v_{c},\lambda_{1},\ldots ,\lambda_{n}\right)=J\left(U,v_{1},\ldots ,v_{c}\right)+\sum_{1}^{n}\lambda_{j}\left(\sum_{i=1}^{c}u_{ij}-1\right)\\ \begin{array}{cccc} & & & =\sum_{i=1}^{c}\sum_{j}^{n}u_{ij}^{m} \end{array}d_{ij}^{2}+\sum_{1}^{n}\lambda_{j}\left(\sum_{i=1}^{c}u_{ij}-1\right) \end{array}$$
where $\lambda_{j}$ is the n constrained Lagrange multiplier of Formula (Equation 8). Then, derive the partial derivatives for all parameters and make their derivatives equal to 0. The updated formula of the membership matrix is as follows:(9)$$v_{i}=\frac{\sum_{j=1}^{n}u_{ij}^{m}x_{j}}{\sum_{j=1}^{n}u_{ij}^{m}}$$
(10)$$u_{ij}=\frac{1}{\sum_{k=1}^{c}{\left(\frac{d_{ij}}{d_{kj}}\right)}^{2/(m-1)}}$$
where i=1,2,…,c, j=1,2,3,…,n. After calculating the fuzzy classification interval of the sample, FCM predicts the sample state $s_{t+1}$ and the maximum $Q_{t+1}$ through the target network. Finally, DQN performs a training learning calculation to obtain the current $Q_{t}=Q\left(s_{t},a_{t}\right)$. Moreover, the target $Q_{t+1}^{*}=r_{t}+Q_{t+1}$ as well as the loss function are calculated according to the current *Q* and the target *Q*.

## 4. Experiments

### 4.1. Dataset

This experiment used the real electricity consumption data of users released by the State Grid Corporation of China (SGCC) as the dataset. Table 1 shows the basic information of this dataset. We randomly divided the dataset into training samples, test samples, and validation samples, at a ratio of 6:2:2. The training samples included normal samples and abnormal samples.

The dataset contained faulty values, wrong measurements, missing values, etc. In this paper, the interpolation method was mainly used to restore a large number of missing values in the electricity dataset. In order to deal with the influence of data of different dimensions on the experimental results, it was necessary to normalize the dataset and the final data between [0, 1]. The data were normalized using the Max-Min scaling method, and the calculation formula is as follows:(11)$$f\left(x_{i}\right)=\frac{x_{i}-\min\left(x\right)}{\max(x)-\min(x)}$$

Here, min(x) denotes the minimum value in the dataset, and max(x) denotes the maximum value in the dataset.

### 4.2. Experimental Environment and Configuration

This experiment was carried out on a server with NVIDIA GeForce GTX 1080 × 2, and the architecture of the model was implemented based on pytorch. The features of the data were regarded as states, the data labels were regarded as actions, and the reward value was represented by a 0/1 function; that is, the reward value was 1 if the classification was correct, and the reward value was 0 if the classification was wrong. Thus, a simulation of the abnormal detection scenario was formed, and the model was trained. Batch size was set to 256, and the learning rate to 0.001. Adam was used to optimize the model.

### 4.3. Evaluation Metrics

In this paper, the accuracy (ACC) was used to evaluate the classification results of the network model. The calculation formula is as follows:(12)$$\mathrm{ACC}=\frac{\mathrm{TP}+\mathrm{TN}}{\mathrm{TP}+\mathrm{TN}+\mathrm{FP}+\mathrm{FN}}$$
where TP represents the number of true positive samples, TN represents the number of true negative samples, FP represents the number of false positive samples, and FN represents the number of false negative samples.

To better evaluate the established abnormal electricity consumption detection model, the evaluation index of the Area Under the Curve (AUC) was adopted, which is usually used as the evaluation index of the two-class detection model. It was mainly used to reflect the specificity and sensitivity of the network model. The AUC value range was generally between 0.5 and 1; the closer it is to 1, the better the classification effect.

### 4.4. Experimental Results and Analysis

#### 4.4.1. Comparison with Existing Methods

This work proposes an FCM state prediction RL network model (FRL) for power abnormal data detection. Table 2 provides the comparison results of the DQN model and the current work using binary classification experiments on the dataset used in this paper. Among them, [26] proposed a method for network traffic detection, which utilized Deep Reinforcement Learning (DRL) to accurately detect intrusion detection system attacks. Reference [27] proposed a Reinforcement Learning (RL) model to analyze feature selection, hyperparameter selection, and intrusion detection problems, which later demonstrated strong experimental validation. Reference [28] proposed a Hierarchically Distributed Fog Computing (HDFC) architecture for deploying a machine learning-based abnormal detection model to detect anomalous data from smart meter sensor data collected by households. Reference [29] proposed a Threshold-based Abnormal Detector (TAD) for energy theft detection in edge data centers. In order to fully prove the validity of the methods, this paper not only compared the methods of smart meter data abnormal detection, but also the network traffic detection methods. By comparing the experimental results, it was found that the direct application of the network traffic abnormal detection method to the abnormal detection task of the electric meter data did not achieve the performance equivalent to that of the electric meter data abnormal detection method; for example, both ACC and AUC were lower. However, other methods of abnormal detection in meter data can achieve better results, such as an ACC and an AUC of 91.2% and 91.4%, respectively. The method in this paper achieved 94.7% performance on ACC and 82.7% performance on AUC, which shows that the DRN model has better performance against unknown attacks.

#### 4.4.2. Parametric Analysis

Figure 2 shows the change curves of the ACC values of the five detection models with epochs. It can be clearly seen that with an increase in the epochs value, the AUC value begins to gradually increase, and then gradually decreases after reaching a certain epoch. By comparing the change curves of the five models, it can be seen that the abnormal electricity consumption detection model of the method in this paper has a good detection effect, and the detection performance is the best when the epoch is set to 60. Therefore, it is not necessary to set the epochs to be as large as possible. When a certain threshold is reached, it may lead to the overfitting of the model. Only by selecting the appropriate epochs can the network model achieve the best detection performance.

Figure 3 shows the detection effect of the method in this paper under different batch_sizes. The batch_size was set to 32, 64, 128, and 256 for the comparative experiments. As can be seen in the figure, when the batch_size was set to 32, the method in this paper could achieve the same AUC value as other batch_sizes with fewer epochs. Different datasets and network models will affect the choice of the batch_size. Therefore, setting an appropriate batch_size can improve the detection performance of the network model and speed up the convergence of the model.

In addition to this, we also analyzed the embedding dimension, the number of layers, and the dropout probability. As shown in Figure 4, the optimal value for the embedding dimension is 128, the optimal value for the dropout probability is 0.5, and the optimal value for the number of layers is 5. When the embedding dimension increases, the model may learn some noise due to overfitting, and the performance decreases. When the number of network layers is increased, the effect of the model continues to improve, which shows that the deep neural network has a better nonlinear feature extraction ability.

As for the comparison of the performance of different classifier models on this dataset, Table 3 shows that the evaluation index results of the proposed method are higher than those of the traditional LR and SVM [1] methods, which indicates that the method in this paper has better performance than the traditional method in the classification of abnormal electricity consumption. The usability of this model in the field of power consumption detection is also shown. By comparing it with the CNN [30], LSTM [31], and TCN [32] deep learning models, it can be seen that the method in this paper has higher accuracy and has advantages in terms of non-technical loss. Compared with general machine learning methods, the proposed model has better prediction performance.

In addition, the detection time is also an important indicator for measuring its performance. Results of the comparison between the proposed method and other methods in terms of abnormal detection time are shown in Figure 5. As can be seen in the Figure, as the number of users increases, the detection time increases; however, the proposed method has the shortest detection time as compared to other methods. Due to the strong feature aggregation ability of the proposed method, the detection time can be shortened.

Based on the experiments above, we find that the proposed method has higher ACC and AUC performance on the same dataset as compared to existing methods, which indicates that the proposed method has higher accuracy when dealing with the problem of power anomaly data detection. In terms of the parameter analysis of the proposed method, we set different number of iterations and batch_sizes. The results show that our method had a significant increase in performance as the number of iterations increased, which indicates the reliability of our method. At the same time, when we analyzed the parameters of the deep learning network, the results show that the proposed enhanced deep neural network had better nonlinear feature extraction ability. When the LR and SVM methods were compared, our method had a large improvement in data prediction accuracy. When comparing the CNN, LSTM, and TCN deep neural network methods, the proposed method had a small improvement in performance, which was due to the better ability of the deep neural network to learn data features. However, deep neural network methods require a large amount of training data to obtain more accurate prediction results. Our method is less dependent on training data as it predicts the state in the next moment. Finally, when analyzing the algorithm’s efficiency, the proposed method can detect data anomalies fastest when facing different numbers of users. Therefore, the proposed method has high stability and reliability in power applications.

## 5. Conclusions

This paper proposed a smart meter data abnormal detection model based on DRL. In this method, the meter data feature was regarded as a state $s_{t}$, and the meter data feature label was regarded as an action $a_{t}$. The method no longer needed to enter the meter data feature $s_{t+1}$, but rather only the electricity meter $a_{t+1}$ data. Meanwhile, this study used the fuzzy c-means method to predict the $s_{t+1}$, which improved the computational accuracy of the Deep Reinforcement Learning model. The experimental results showed that the proposed method can effectively improve the ACC and AUC performance of anomaly detection as compared with some recent studies using the same dataset. Furthermore, the results showed that our method requires the least amount of time to identify unknown attacks.

## Figures and Tables

**Figure 1 sensors-22-08543-f001:**
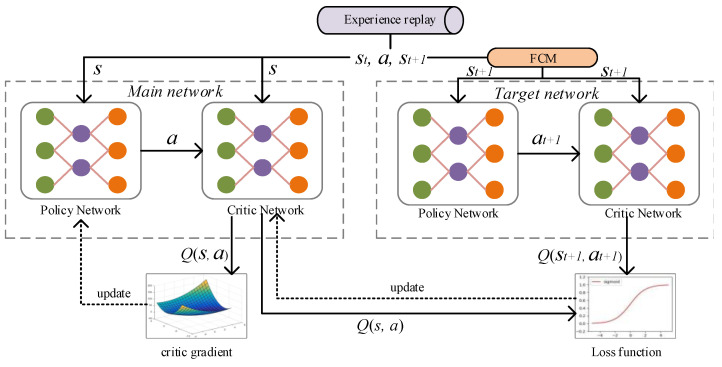
The training process of the DQN network model.

**Figure 2 sensors-22-08543-f002:**
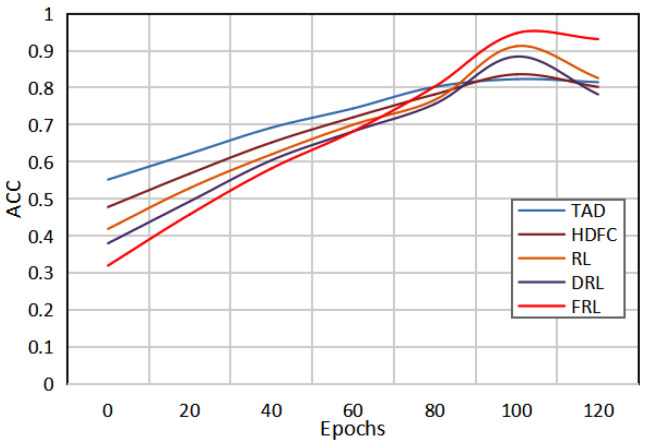
Comparison of the results of the five methods under different epochs.

**Figure 3 sensors-22-08543-f003:**
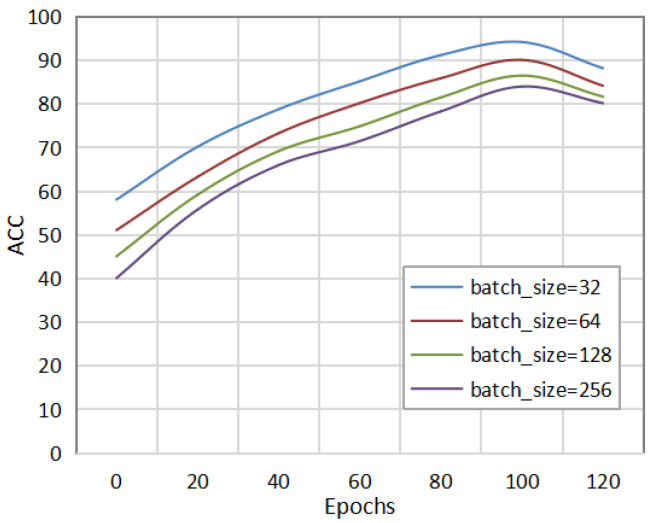
Comparison of experimental results under different batch_sizes.

**Figure 4 sensors-22-08543-f004:**
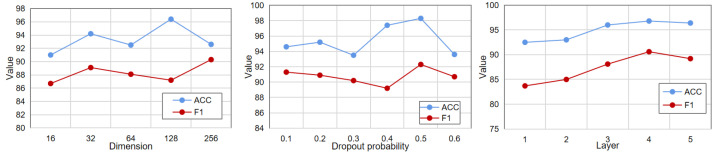
Parametric analysis.

**Figure 5 sensors-22-08543-f005:**
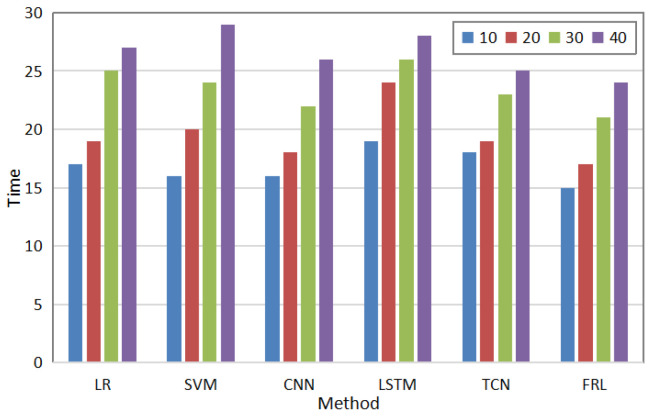
Comparison of detection times of different methods.

**Table 1 sensors-22-08543-t001:** Basic information of the dataset.

User Information	Value
Time range	1 January 2014–7 February 2017
Total number of samples	149,186
Number of normal samples	140,434
Number of abnormal samples	8752
Number of training samples	89,500
Number of validation samples	29,843
Number of test samples	29,843

**Table 2 sensors-22-08543-t002:** Comparison of experimental results.

Methods	ACC	AUC
DRL [26]	0.823	0.690
RL [27]	0.856	0.724
HDFC [28]	0.912	0.728
TAD [29]	0.914	0.746
FRL	0.947	0.827

**Table 3 sensors-22-08543-t003:** Comparison of experimental results under different classifiers.

Classifier	ACC	AUC
LR	0.582	0.167
SVM [1]	0.534	0.351
CNN [30]	0.823	0.860
LSTM [31]	0.693	0.824
TCN [32]	0.825	0.836
FRL	0.882	0.879

## Data Availability

The data presented in this paper will be made available on request via the corresponding author’s email with appropriate justification.

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
