# Peer review of "Deep Reinforcement Learning for the Detection of Abnormal Data in Smart Meters"

_sensors, 2022, doi:10.3390/s22218543_

Round 1
Reviewer 1 Report
The autor show a Deep Reinforcement Learning Method for the Detection of Abnormal Data in Smart Meters.
The manuscript is well organized and presented.
Some suggestion to improve the manuscript are:
1. I am not sure if the first section is 0
2. Table 2 must be contain the method applied in each related work
3. It is necessary a more detailed results discussion
Author Response
Point 1: I am not sure if the first section is 0
Response 1: Thank you very much for your suggestion. We have revised in the latest manuscript that the first section is 1.
Point 2: Table 2 must be contain the method applied in each related work
Response 2: Thank you very much for your suggestion. We have revised this question in the latest manuscript.
Point 3: It is necessary a more detailed results discussion
Response 3: Thank you very much for your opinion. We have refined the results discussion in the latest manuscript and fully illustrate our work. Modifications have been highlighted.
Reviewer 2 Report
The authors conducted a very interesting study. The article can be significantly improved and will be better perceived by readers. Please see my comments in the attached file.

Author Response
Point 1: Line 9: The use of abbreviations in the annotation is not allowed
Response 1: Thank you very much for your suggestion. We have removed this abbreviation.
Point 2: Line 15: in this section there is no clear statement of the research problem, it is not clear what exactly is the problem for which this research was conducted
Response 2: Thank you very much for your suggestion. We have revised the problem description in this paragraph. The main purpose of this work is to solve the typical problem of smart meter data anomaly detection based on artificial intelligence algorithm model. The causes of this typical problem include tampering with meter data, network attacks, and abnormal operation of the meter. We aim to design a reinforcement learning model at the edge of the electricity meter, which can predict the electricity consumption according to the user's electricity consumption habits, and then judge the abnormal electricity consumption. It can greatly reduce the pressure of cloud computing.
Point 3: Line 30-38: I do not agree with any of the statements in this paragraph. Maybe the authors had in mind some special method, which is called cloud computing, which I do not know about?
Response 3: Thank you very much for your opinion. We have removed this description of cloud computing.
Point 4: Line 61: no explanation what the Q function is
Response 4: Thanks for your comment. We have added a description of the Q function in the latest manuscript.
Point 5: Line 75: this section lists existing methods, but does not show exactly how they relate to this study and why they are not applicable in this case.
Response 5: Thank you for your suggestion. We try to introduce the current data security issues in smart grids and related research in Section 2.1. Including the use of related personnel has predicted power data anomalies with K-means and particle swarm methods. Section 2.2 addresses the typical problem of anomaly detection in smart grids, which is the focus of this work. At the same time, some mainstream machine learning solutions are described in Section 2.2. Section 2.3 mainly highlights the development of reinforcement learning, which also introduces the proposed purpose of our method. At the same time, we have included some clarifications in the new manuscript in response to the reviewers' questions.
Point 6: Line 76: It is not clear for what purpose this subparagraph is written.
Response 6: Thank you for your suggestion. The main purpose of this section is to describe the current application of artificial intelligence algorithms in smart grids. We attempt to illustrate research implications in this section by presenting the work of others. For example, we would like to illustrate how to ensure information security when a smart grid system is attacked during operation through References 7 and 8. References 9 and 10 are used to illustrate that some malicious attacks can be countered by distributed services and encryption algorithms. Reference 11 is used to illustrate that methods such as K-means and particle swarms can be combined to solve the problem of data anomaly detection.
Point 7: Line 181: from the foregoing, it is absolutely not clear which factors influence the process and how many of them. The word "many" doesn't make sense.
Response 7: Thanks a lot for your opinion. We have revised this question in the latest manuscript.
Point 8: Line 191: what is this equation? it has been mentioned before
Response 8: Thank you very much for your suggestion. We have fixed this misrepresentation.
Point 9: Line 202-Equation 1: here and below, none of the symbols used in the formulas is deciphered anywhere in the article, this is very inconvenient for reading and understanding what is written
Response 9: Thanks a lot for your opinion. We have added a description of Equation 1, including s and a, in the latest manuscript.
Point 10: Table 1 Basic information of the dataset: not enough information about the dataset. It is not clear what data was used to train the model and what data to test it.
Response 10: Thank you for your suggestion. We have added relevant clarifications to the latest manuscript. We randomly divide the dataset into training samples, test samples and validation samples, and their ratio is 6:2:2. The training samples include normal samples and abnormal samples.
Point 11: Line 283 section 3.4: comparisons are unclear. Why were these methods chosen for comparison? Were the comparisons in Table 2 performed on the same dataset as proposed in this study? No clear criteria for comparison
Response 11: Thank you for your suggestion. We state this note about the dataset and related work in our latest manuscript. These studies in Table 2 are similar to those in this paper, so we choose the algorithm of the work as a comparison. During this experiment, we chose the same dataset as shown in Table 1 based on different methods. The comparison criteria for our study are two parameters, ACC and AUC, which have been introduced in Section 3.3.
Point 12: Line 321-Figure 3: comparisons are unclear. Why were these methods chosen for comparison? Were the comparisons in Table 2 performed on the same dataset as proposed in this study? No clear criteria for comparison
Response 12: Thank you for your suggestion. We have revised this question as indicated in the previous reply.
Point 13: Line 343: The conclusions should be expanded. They do not reflect all the results of the study
Response 13: Thank you very much for your suggestion. We have refined our conclusions in the latest manuscript and fully illustrate our work.
Reviewer 3 Report
The abstract needs to be rewritten.
Please don't use "our" in the table and graphs. Try to suggest some names for your work.
As in this paper, the authors use Q-learning and Fuzzy_C methods. Could you please explain how your work is different? These methods are already proposed. Did you make some changes to these methods or use them?
Line 224. after many iterations, please explain properly. many could be 5,10,100,1000,etc
Line 288, Reference 288 [27] mainly uses RL; Please rewrite it. (at many other occasions, you used the same style)
Author Response
Point 1: The abstract needs to be rewritten.
Response 1: Thank you very much for your suggestion. We have rewritten the Abstract in the latest manuscript.
Point 2: Please don't use "our" in the table and graphs. Try to suggest some names for your work.
Response 2: Thank you very much for your suggestion. We have revised this question in the latest manuscript. We replaced "our" in the table and graphs with FRL.
Point 3: As in this paper, the authors use Q-learning and Fuzzy_C methods. Could you please explain how your work is different? These methods are already proposed. Did you make some changes to these methods or use them?
Response 3: Thank you very much for your suggestion. The traditional Q-learning method uses the policy function and prior knowledge to predict the state at the next moment, which is an unsupervised learning method. In order to take advantage of model update in deep reinforcement learning, we use deep Q-learning method to better learn the feature changes of power data. However, in order to be able to obtain better prediction accuracy, we try to use historical datasets to guide the parameter update of this unsupervised learning model. Therefore, we add a target network and a state network composed of a deep learning model to the deep Q-learning network. At the same time, in order to find samples with high weight value from more historical samples as the state input of Q-learning. We try to use the FCM method to achieve a preliminary state prediction in order to achieve the purpose of optimizing the input of the target network, so as to obtain a more robust prediction model. In general, we have made an organic combination to take advantage of the two algorithms to achieve the purpose of efficient anomaly detection in power data.
Point 4: Line 224. after many iterations, please explain properly. many could be 5,10,100,1000,etc
Response 4: thank you very much for your suggestion. We have explained the number of iterations in the latest manuscript, and we set the network's epoch=100 on the training of this deep network.
Point 5: Line 288, Reference [27] mainly uses RL; Please rewrite it. (at many other occasions, you used the same style)
Response 5: Thank you very much for your suggestion. We have revised this question in the new manuscript.
Round 2
Reviewer 3 Report
No more comments.